# S-GRPO: Early Exit via Reinforcement Learning in Reasoning Models

**Muzhi Dai**[1]*, **Chenxu Yang**[2]*, **Qingyi Si**[1]†,

[1]Huawei Technologies Co., Ltd.

[2]Institute of Information Engineering, Chinese Academy of Sciences, Beijing, China

mzdai666@gmail.com, yangchenxu@iie.ac.cn, siqingyi@huawei.com

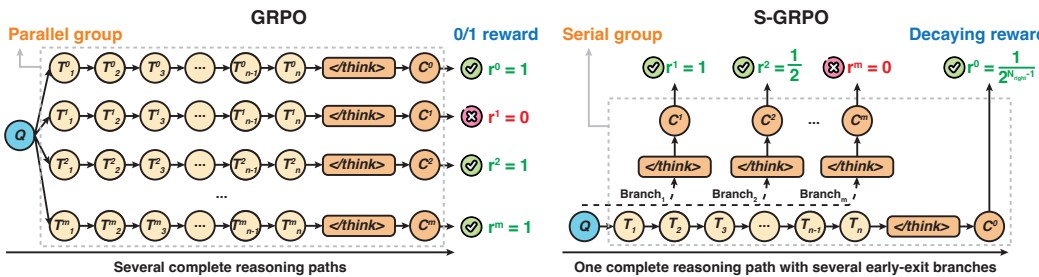

Figure 1: Comparison of parallel-group-relative GRPO and our serial-group-relative S-GRPO.

## Abstract

As Test-Time Scaling emerges as an active research focus in the large language model community, advanced post-training methods increasingly emphasize extending chain-of-thought (CoT) generation length, thereby enhancing reasoning capabilities to approach Deepseek R1-like reasoning models. However, recent studies reveal that reasoning models (even Qwen3) consistently exhibit excessive thought redundancy in CoT generation. This overthinking issue arises from the inherent limitations of conventional outcome-reward reinforcement learning, which systematically overlooks the regulation of intermediate reasoning processes. This paper introduces **S**erial-**G**roup Decaying-**R**eward **P**olicy **O**ptimization (S-GRPO), a novel reinforcement learning paradigm that enables models to implicitly evaluate the sufficiency of intermediate reasoning steps, thereby facilitating early exit in CoT generation. Unlike GRPO, which samples multiple possible reasoning paths in parallel (parallel group), S-GRPO only samples one reasoning path and serially selects multiple temporal positions from the path to exit thinking and directly generate answers (serial group). For correct answers within a serial group, rewards gradually decrease based on the exit positions along the reasoning path from front to back. This design encourages the model to produce more accurate and concise thoughts, while also incentivizing early thinking termination when appropriate. Empirical evaluations demonstrate that S-GRPO is compatible with state-of-the-art reasoning models, including Qwen3 and Deepseek-distill. Across diverse benchmarks such as GSM8K, AIME 2024, AMC 2023, MATH-500, and GPQA Diamond, S-GRPO achieves a substantial reduction in sequence length (40.4%~61.1%) while simultaneously improving accuracy (absolute 0.72%~3.92%).

---

\*    Equal contribution. The work was done when Muzhi and Chenxu were interns at Huawei.
†    Corresponding Author.

# 1 Introduction

Test-Time Scaling [1] demonstrates a strong correlation between extended chain-of-thought (CoT) and enhanced reasoning capabilities in Large Language Models, which is more effective than scaling model parameters [2]. The success of DeepSeek-R1 [3] and GPT-o1 [4] has further sparked the research interest in reasoning models [5] within the LLM community. Furthermore, reinforcement learning [6, 7, 8, 9, 10] (RL) in post-training has demonstrated significant potential in stimulating long chain-of-thought generation and strengthening deep-thinking capabilities.

However, recent studies [11, 12] have identified a critical limitation in reasoning models: their tendency to engage in redundant thought processes, a phenomenon referred to as Overthinking [11, 13]. They frequently generate unnecessarily lengthy reasoning sequences [13, 14], including irrelevant information and superfluous reasoning steps. This redundant thought inflates computational overhead and even undermines reasoning accuracy by diverting the model from valid reasoning pathways to incorrect ones [15]. We attribute this issue to the inherent limitations of 0/1 outcome-reward RL (e.g., GRPO [6]), where reliance on final outcome rewards fails to identify when intermediate reasoning steps are sufficient.

As shown in Figure 1 (left), a standard outcome-reward RL, such as GRPO, tends to sample a query multiple times in parallel to obtain a parallel group. All reasoning chains and corresponding conclusions within the parallel group receive 0/1 outcome rewards, which reinforces correct outcomes but overlooks the presence of overthinking or inefficiencies in intermediate reasoning steps. While this approach successfully aggregates pass@k reasoning capability into pass@1, the neglect of regulating intermediate reasoning results in inefficient inference. Conversely, as illustrated in Figure 1 (right), we construct a serial group by sequentially generating multiple completions during a single CoT process. By prioritizing rewards for earlier correct completions, this approach encourages models to demonstrate complete reasoning capabilities in the initial phases, enabling early exit and preventing overthinking.

Motivated by this, we propose **S**erial-**G**roup Decaying-**R**eward **P**olicy **O**ptimization (**S-GRPO**), a simple yet effective modification to address standard outcome-reward RL (GRPO)'s inability to regulate intermediate reasoning processes. During S-GRPO training, we construct a serial group for each query using a two-phase rollout process. In the first phase, a complete reasoning path is generated. In the second phase, subsequent rollouts introduce early-exit interventions at different positions along the reasoning path generated in the first phase, producing intermediate answers. Finally, the serial group is formed by combining the complete reasoning path from the first phase with the intermediate answers appended to its corresponding truncated reasoning path. On this basis, we assign the rewards that decay according to their order of early exit for the correct ones within the serial group. S-GRPO culminates in computing serial-group relative advantages and using their policy gradient to update model parameters.

Our contributions are threefold:

- We pioneer a serial-group RL paradigm that overcomes the critical limitation of outcome-reward RL in regulating intermediate reasoning processes, accompanied by an open-sourced training framework (released once accepted).
- The proposed S-GRPO algorithm enables models to produce higher-quality reasoning paths during the early stages of CoT generation, and implicitly early-exit once sufficiency is achieved. S-GRPO preserves the integrity of the original reasoning process via a two-stage rollout procedure, ensuring that the model's pre-existing reasoning abilities are not compromised, which is well-suited as the final stage of post-training.
- Extensive experiments across GSM8K [16], AIME 2024 [17], AMC 2023 [18], MATH-500 [19], and GPQA [20] benchmarks with Qwen3 [21] and Deepseek-series reasoning models, demonstrate 0.72%~3.92% absolute accuracy improvement alongside 40.4%~61.1% average token reduction, establishing an efficiency-accuracy synergistic improvement.

# 2 Method

The proposed **Serial-Group Decaying-Reward Policy Optimization (S-GRPO)** is a novel reinforcement learning mechanism that innovatively leverages rule-based outcome rewards to regulate

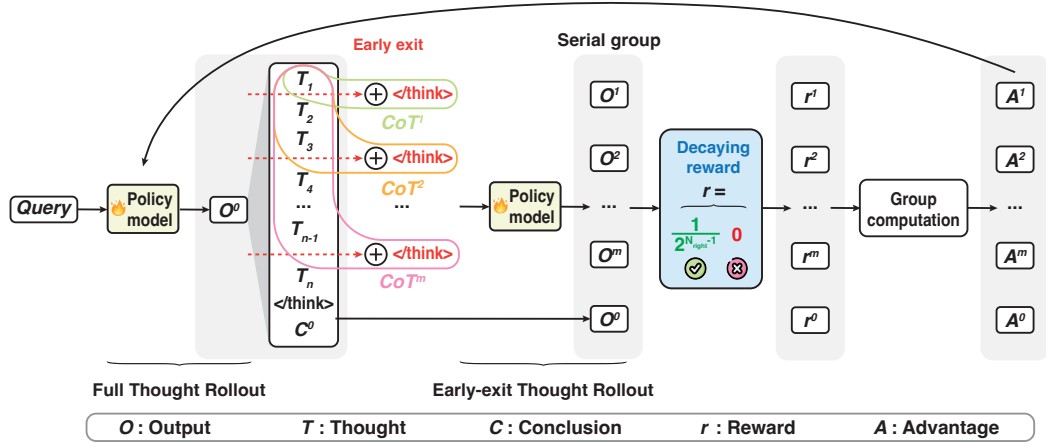

Figure 2: The framework of S-GRPO. The complete answer inducer is omitted in the figure and is represented by </think> instead. The complete answer inducer is "`Time is limited, stop thinking and start answering.\n</think>\n\n`"

intermediate reasoning processes and improve reasoning efficiency. During S-GRPO training, the LLM exits early and generates answers at different positions within a complete CoT, forming a serial group. Rewards for correct answers are assigned based on position, with earlier thinking exits receiving relatively higher rewards. This strategy encourages the model to produce high-quality reasoning earlier and terminate once sufficient reasoning is achieved. In particular, the framework is divided into three stages: Serial-Group Generation, Decaying Reward Strategy, and Advantage Computation and Parameter Update.

## 2.1 Serial-Group Generation

GRPO is originally designed to enable reasoning models to achieve their pass@k potential under pass@1 settings. To achieve this, GRPO generates multiple CoTs in parallel for each query, forming a parallel group, and rewards only those with correct answers. In contrast, S-GRPO aims to achieve efficient reasoning through sufficient-reasoning early exit. Specifically, we perform early-exit interventions at different positions within a single CoT to construct a serial group, thereby allowing the RL training to compare the thought sufficiency in different positions along the reasoning path. It consists of two stages: Full Thought Rollout and Early-exit Thought Rollout.

### 2.1.1 Full Thought Rollout

In the Full Thought Rollout stage, the model generates a complete reasoning path sequentially for each query, represented as $\{O^0 = T_1, T_2, \ldots, T_n, </think>, C_0\}$. To expose the model to diverse reasoning sufficiency scenarios and enhance its ability to handle early exits effectively, we adopt **random-length truncation** during training. Specifically, the reasoning sequence is truncated at $m$ randomly selected temporal positions $P_i$, where $P_i = T_i$ $and$ $i \sim \text{Uniform}(1, n)$, ensuring that the truncation points are uniformly distributed across the reasoning path. This randomness allows the model to implicitly evaluate reasoning sufficiency across varying depths and prevents overfitting to specific reasoning path lengths.

### 2.1.2 Early-exit Thought Rollout

In the Early-exit Thought Rollout stage, the policy model extracts several early-exit reasoning paths $\{CoT^1, CoT^2, ..., CoT^m\} = \{O^0[: P_1], O^0[: P_2]..., O^0[: P_m]\}$, representing truncated segments of the full reasoning path at different positions. For each early-exit path, the model generates corresponding answers $C_1, C_2, \ldots, C_m$.

In particular, we insert the prompt "`Time is limited, stop thinking and start answering. \n</think>\n\n`" at each truncated position $P_i$. This prompt explicitly in-

structs the model to halt further reasoning and produce answers $C^i$. Details of this process are illustrated in Figure 5.

To ensure that the serial group contains early exit samples with correct answers (obtained by Early-exit Thought Rollout), we sample more queries than the training batch required (over-sampling) for data filtering, such as DAPO [22].

## 2.2 Decaying Reward Strategy

To encourage models to produce adequate and correct reasoning steps at earlier stages of CoT generation for accurate problem-solving, we propose a Decaying Reward Strategy. This mechanism assigns rewards based on the correctness of answers generated during the two-time rollouts ( $C_1, C_2, \ldots, C_m, and\ C_0$ ), while decaying rewards according to their order of early exits. For the answer $C^i$ of each response $O^i$, the reward $r^i$ is defined as follows:

$$r^i = \begin{cases} \frac{1}{2^{N_{\text{right}}-1}}, & \text{if } C^i \text{ is correct,} \\ 0, & \text{if } C^i \text{ is incorrect.} \end{cases} \tag{1}$$

Where $N_{\text{right}}$ refers to the accumulated number of correct answers up to and including the current position. The Decaying Reward Strategy is designed with dual objectives: (1) Exponential decay for correct answers: The strategy applies exponentially diminishing rewards to enhance the quality of earlier reasoning steps, which is overlooked by binary 0/1 outcome rewards. (2) Zero reward for incorrect answers: The strategy enforces a correctness-first optimization strategy, ensuring the model maintains robust core reasoning capabilities.

This dual-objective design strikes a balance between reasoning sufficiency and efficiency, guiding the model to produce reasoning sequences that are both accurate and concise.

## 2.3 Advantage Computation and Parameter Update

After computing the decaying rewards, S-GRPO calculates the advantage for each response in the serial group. Specifically, the advantage is calculated by subtracting the mean reward of the group from its corresponding reward, as defined by the formula: $\hat{A}_i = r_i - \text{mean}(r_i)$. Here, for training stability, the standard deviation is removed from the advantage computation compared to the GRPO. Subsequently, the computed advantage for each sample is broadcast to all corresponding response tokens. Finally, parameter updates are performed based on the advantage values of each sample (Algorithm 1). The optimization objective is as follows:

$$\mathcal{J}_{\text{S-GRPO}}(\theta) = \mathbb{E}_{[q \sim P(Q), \{o_i\}_{i=1}^G \sim \pi_{\theta_{\text{old}}}(O|q)]}$$

$$[\frac{1}{G} \sum_{i=1}^{G} \frac{1}{|o_i|} \sum_{t=1}^{|o_i|} \{\min[\frac{\pi_\theta^{i,t}}{\pi_{\theta_{\text{old}}}^{i,t}} \hat{A}_{i,t}, \ \text{clip}(\frac{\pi_\theta^{i,t}}{\pi_{\theta_{\text{old}}}^{i,t}}, \ 1-\epsilon, \ 1+\epsilon)\hat{A}_{i,t}]\}] \tag{2}$$

where $\pi^{i,t} = \pi(o_{i,t} \mid q, o_{i,<t})$ denotes the conditional probability of the token at position $t$. The models $\pi_{\theta_d}, \pi_{\theta_{d\text{old}}}$ correspond to the training model and sampling model, respectively. $q$ represents the input query, and $\{o_i\}_{i=1}^G$ are the full and early-exit thought rollout responses generated by the model. The advantage $\hat{A}_i$ for each response is computed as $\hat{A}_i = r_i - \text{mean}(r_i)$, where $r_i$ is the decaying reward assigned to the response. For token-level advantage $\hat{A}_{i,t}$, it is defined to be equal to the corresponding sequence-level advantage $\hat{A}_i$. The hyperparameter $\epsilon$ is used to bound the importance sampling ratio.

# 3 Experiments

## 3.1 Experimental Setup

**Training datasets.** We selected problems from DeepMath-103K [23] to build our training set. This dataset is a large-scale and challenging collection of mathematics problems, focusing on difficulty levels ranging from grade 5 to grade 10. It addresses the lack of sufficient complexity present in existing datasets by offering approximately 103,000 carefully curated problems. The dataset is

**Algorithm 1** Serial-Group Decaying-Reward Policy Optimization (S-GRPO)

---
**Require:** Query $Q$, Policy model $\pi_\theta$, Number of positions to sample $m$
**Ensure:** Updated policy parameters $\theta$
 1: **Full Thought Rollout:**
 2: Generate complete reasoning path $O^0 = (T_1, T_2, \ldots, T_n, \text{</think>}, C^0)$ using $\pi_\theta$ for query $Q$
 3: Sample $m$ temporal positions $P_i \in \{P_1, P_2, \ldots, P_m\}$ where $P_i \sim \text{Uniform}(1, n)$
 4: **Early-exit Thought Rollout:**
 5: $T_{\text{</think>}}$ = tokenizer.encode("Time is limited, stop thinking and start answering. \n</think>\n\n")
 6: **for** each position $P_i$ **do**
 7:     Append $T_{\text{</think>}}$ after position $P_i$ to form $CoT^i = (T_1, T_2, \ldots, T_{P_i}, T_{\text{</think>}})$
 8:     Generate answer $C^i$ using policy model $\pi_\theta$ conditioned on $CoT^i$
 9:     Form output $O^i = (CoT^i, C^i)$
10: **end for**
11: **Rule-based Dynamic Decaying-reward:**
12: $N_{\text{right}} \leftarrow 0$                                                    ▷ Counter for correct answers
13: **for** each output $O^i$ where $i \in \{1, 2, \ldots, m, 0\}$ in order of position **do**
14:     **if** $C^i$ is correct **then**
15:         $N_{\text{right}} \leftarrow N_{\text{right}} + 1$
16:         $r^i \leftarrow \frac{1}{2^{N_{\text{right}}-1}}$                               ▷ Exponentially decaying reward
17:     **else**
18:         $r^i \leftarrow 0$                                                  ▷ Zero reward for incorrect answers
19:     **end if**
20: **end for**
21: **Group Computation:**
22: Compute advantages $A^i$ for each output based on rewards $r^i$
23: Update policy parameters $\theta$ using policy gradient with advantages $A^i$

---

constructed through an extensive data cleaning process applied to known benchmarks, ensuring no overlap with widely used benchmarks.

**Benchmarks.** To comprehensively assess the models' capabilities across a range of reasoning tasks, we select five popular benchmarks that reflect diverse levels of difficulty: **GSM8K** [16] is a well-curated collection of 1,319 problems in elementary mathematics. This benchmark is specifically designed to evaluate multi-step reasoning in foundational math tasks. Problems typically involve two to eight sequential operations, relying primarily on basic arithmetic performed over multiple intermediate steps. **AIME 2024** [17] includes 30 advanced problems selected from the 2024 edition of the American Invitational Mathematics Examination. This highly regarded competition tests participants' ability to reason mathematically across a broad range of topics, including arithmetic, algebra, combinatorics, geometry, number theory, and probability—core components of secondary-level mathematics. **AMC 2023** [18] consists of 40 questions spanning key areas such as algebra, geometry, number theory, and combinatorics. As part of the American Mathematics Competitions organized by the Mathematical Association of America (MAA), AMC aims to foster problem-solving skills and identify promising young mathematicians. **MATH-500** [19] presents a set of challenging problems drawn from high school-level mathematical competitions. To ensure comparability with prior studies, we use the same subset of 500 problems originally compiled by OpenAI for evaluation purposes [24]. In addition to these mathematical evaluations, we also examine performance on scientific reasoning tasks, using the following benchmark: **GPQA** [20] is a rigorously constructed dataset containing graduate-level questions in physics, chemistry, and biology. Notably, even domain experts with PhDs achieve only 69.7% accuracy on this benchmark [25]. For our experiments, we focus on the highest-quality subset, **GPQA Diamond**, which contains 198 carefully chosen questions.

**Baselines.** We compare S-GRPO with various existing efficient reasoning methods, including the training-free, output-based approach DEER [15], the off-policy optimization method ConCISE [26], and on-policy RL-based approaches such as original GRPO, RL + Length Penalty [27], and ShorterBetter [28]. DEER makes early-exit decisions during the inference phase, guided by the confidence scores of intermediate answers. ConCISE generates more concise chains of thought by

Table 1: Experimental results on four large reasoning models.

| Method | GSM8K | | AIME 2024 | | AMC 2023 | | MATH-500 | | GPQA | | Overall | |
|---|---|---|---|---|---|---|---|---|---|---|---|---|
| | Acc | Tokens | Acc | Tokens | Acc | Tokens | Acc | Tokens | Acc | Tokens | Acc | Tokens |
| *DeepSeek-R1-Distill-Qwen-7B* | | | | | | | | | | | | |
| *Vanilla* | 92.4 | 1,833 | 55.4 | 13,232 | 77.2 | 9,693 | 85.8 | 5,590 | 50.1 | 15,385 | 72.18 | 9,147 |
| *DEER* | 88.8 | 917 | 53.3 | 10,971 | 87.5 | 4,142 | 91.8 | 2,431 | 47.5 | 5,280 | $73.78_{+1.60}$ | $4,748_{-48.1\%}$ |
| *ConCISE$_{SFT}$* | 92.9 | 832 | 52.1 | 9,751 | – | – | 92.0 | 2,244 | 50.0 | 5,892 | – | – |
| *ConCISE$_{SimPO}$* | 92.1 | 715 | 48.3 | 7,745 | – | – | 91.0 | 1,946 | 48.0 | 4,859 | – | – |
| *GRPO* | 93.2 | 1,767 | 55.0 | 13,451 | 87.5 | 9,887 | 93.6 | 5,317 | 50.7 | 15,817 | $76.00_{+3.82}$ | $9,248_{+1.1\%}$ |
| *RL + Length Penalty* | 92.4 | 1,062 | 51.9 | 7,464 | 86.9 | 3,540 | 92.2 | 2,451 | 49.1 | 3,984 | $74.50_{+2.32}$ | $3,700_{-59.5\%}$ |
| *ShorterBetter* | – | – | 53.3 | 5,288 | 75.9 | 2,580 | – | – | – | – | – | – |
| *S-GRPO* | 93.8 | 906 | 56.0 | 7,377 | 87.5 | 3,494 | 92.4 | 2,252 | 50.8 | 3,751 | **$76.10_{+3.92}$** | $3,556_{-61.1\%}$ |
| *DeepSeek-R1-Distill-Qwen-14B* | | | | | | | | | | | | |
| *Vanilla* | 94.2 | 2,129 | 64.4 | 11,099 | 90.5 | 5,527 | 93.5 | 3,844 | 59.2 | 6,034 | 80.36 | 5,727 |
| *DEER* | 93.3 | 982 | 70.0 | 10,335 | 90.0 | 4,349 | 91.4 | 2,753 | 57.1 | 4,767 | $80.36_{+0.0}$ | $4,637_{-19.0\%}$ |
| *GRPO* | 95.3 | 2,120 | 65.8 | 13,504 | 91.9 | 6,595 | 94.0 | 4,471 | 58.9 | 7,354 | $81.18_{+0.82}$ | $6,809_{+18.9\%}$ |
| *RL + Length Penalty* | 94.7 | 775 | 55.0 | 7,950 | 88.1 | 3,396 | 92.4 | 1,993 | 56.0 | 4,380 | $77.24_{-3.12}$ | $3,699_{-35.4\%}$ |
| *S-GRPO* | 96.2 | 724 | 64.4 | 6,712 | 91.9 | 3,352 | 93.6 | 2,146 | 59.3 | 3,334 | **$81.08_{+0.72}$** | $3,254_{-43.2\%}$ |
| *Qwen3-8B* | | | | | | | | | | | | |
| *Vanilla* | 95.4 | 2,370 | 74.1 | 15,326 | 91.3 | 9,452 | 93.4 | 5,577 | 55.6 | 8,741 | 81.96 | 8,293 |
| *DEER* | 95.5 | 981 | 76.7 | 11,287 | 95.0 | 6,198 | 93.4 | 3,208 | 52.5 | 3,104 | $82.62_{+0.66}$ | $4,956_{-40.2\%}$ |
| *GRPO* | 95.8 | 2,355 | 72.7 | 15,154 | 92.8 | 8,983 | 94.4 | 5,440 | 55.8 | 8,819 | $82.30_{+0.34}$ | $8,150_{-1.7\%}$ |
| *RL + Length Penalty* | 95.4 | 1,323 | 73.8 | 9,666 | 93.4 | 5,042 | 94.2 | 3,247 | 56.2 | 5,293 | $82.60_{+0.64}$ | $4,914_{-40.7\%}$ |
| *S-GRPO* | 96.1 | 1,292 | 77.3 | 8,810 | 95.0 | 5,962 | 95.2 | 3,166 | 57.7 | 5,271 | **$84.26_{+2.3}$** | $4,900_{-40.9\%}$ |
| *Qwen3-14B* | | | | | | | | | | | | |
| *Vanilla* | 95.5 | 1,909 | 75.4 | 14,116 | 96.9 | 7,576 | 95.2 | 5,078 | 58.8 | 7,576 | 84.36 | 7,251 |
| *DEER* | 95.5 | 908 | 76.7 | 10,333 | 95.0 | 5,099 | 94.8 | 2,987 | 57.1 | 2,435 | $83.82_{-0.54}$ | $4,352_{-40.0\%}$ |
| *GRPO* | 96.1 | 1,956 | 77.7 | 14,544 | 98.4 | 8,000 | 95.8 | 5,140 | 59.3 | 7,966 | $85.46_{+1.1}$ | $7,521_{+3.7\%}$ |
| *RL + Length Penalty* | 95.8 | 1,090 | 74.8 | 9,056 | 96.6 | 5,059 | 95.8 | 2,866 | 59.4 | 4,949 | $84.48_{+0.12}$ | $4,604_{-36.5\%}$ |
| *S-GRPO* | 96.3 | 952 | 77.9 | 8,932 | 97.8 | 4,537 | 96.4 | 2,652 | 60.6 | 4,537 | **$85.80_{+1.44}$** | $4,322_{-40.4\%}$ |

integrating specialized prompt tokens and applying early-exit mechanisms during inference, followed by SFT/SimPO to further encourage succinct reasoning. RL + Length Penalty assigns reward values based on the deviation of each correct response length from the mean, penalizing longer correct responses. ShorterBetter promotes shorter yet accurate reasoning paths by assigning higher rewards to compact chains that yield correct answers based on GRPO.

**Models.** We evaluate S-GRPO on four large reasoning models, including DeepSeek-R1-Distill-Qwen-7B, DeepSeek-R1-Distill-Qwen-14B [3], Qwen3-8B, and Qwen3-14B [21]. Despite achieving state-of-the-art performance on reasoning tasks, these models tend to generate overly verbose reasoning processes that contain excessive redundant information.

**Metrics.** S-GRPO is designed to improve correctness while minimizing inference length, thereby enabling more efficient reasoning. To evaluate its performance, we adopted two key metrics: *Accuracy* (i.e., pass@1) and *Token Count* (Tokens). Due to the inherent instability of generating long sequences in reasoning models and the limited sample sizes of certain benchmarks, we conducted multiple evaluation runs and reported the averaged results in the tables. Specifically, we performed 16 trials on AIME 2024 and AMC 2023, 8 trials on MATH-500 and GPQA Diamond, and 4 trials on GSM8K.

**Training details.** For S-GRPO, we use a learning rate of $1 \times 10^{-6}$ and randomly select 8 temporal positions for each query. Since we adopt an on-policy mode, the generation batch size and training batch size are both set to $128 \times 8$. For GRPO, we use the same learning rate and batch size settings. For RL + Length Penalty, we follow the settings described in its original paper [27] and set the scalar parameter $\alpha$ to 0.2. Across all experiments, we employ Adam [29] as the standard optimizer.

## 3.2 Experimental Results

**Main results.** The experimental results in Table 1 demonstrate that S-GRPO consistently outperforms existing baselines across five benchmark datasets and four reasoning models, achieving significant improvements. Compared to vanilla reasoning models, S-GRPO achieves an average accuracy (absolute) improvement of 0.72% to 3.92%, while reducing the generated sequence length by 40.4% to 61.1%. S-GRPO achieves notable improvements on in-domain mathematical reasoning benchmarks (e.g., GSM8K, AIME 2024, AMC 2023, MATH-500) and out-of-domain scientific reasoning tasks (e.g., GPQA), demonstrating its effectiveness and robustness. Specifically, when applied to DeepSeek-R1-Distill-Qwen-7B, S-GRPO achieves a 10.3-point increase in accuracy on

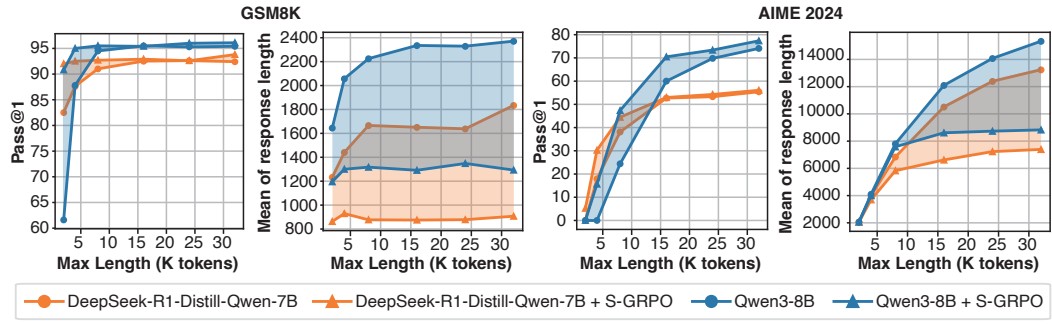

Figure 3: Performance of DeepSeek-R1-Distill-Qwen-7B and Qwen3-8B without or with S-GRPO training on GSM8K and AIME 2024 under different generation-length budgets.

AMC 2023 while utilizing only 36% of the original reasoning budget. Similarly, it obtains a 0.6-point improvement on AIME 2024 with just 56% of the original inference length.

**Comparison with SOTAs.** We compare S-GRPO with several approaches designed for efficient reasoning. For example, DEER reduces reasoning length by only 7%∼27% on AIME 2024 and sometimes sacrifices accuracy, and its effectiveness is mainly limited to simple tasks. In contrast, S-GRPO consistently reduces reasoning length across both simple and complex tasks while simultaneously improving reasoning accuracy. Compared to the original GRPO, S-GRPO achieves comparable or even better accuracy while significantly shortening the inference trajectory, indicating that the proposed Serial-Group Generation mechanism does not hinder the model's exploration capability in reinforcement learning. Moreover, the reward shaping based on serialized intermediate outputs more effectively guides the model toward efficient reasoning.

Compared with the most recent training-based efficient reasoning methods, S-GRPO achieves the best in both reasoning length reduction and accuracy. This can be explained by the fact that ConCISE's off-policy optimization forces the model to fit into a new data distribution, and ShorterBetter and RL + Length Penalty overly emphasize length reduction, at the expense of generalization and accuracy-driven optimization. In contrast, S-GRPO preserves the integrity of the original reasoning process via a two-stage rollout procedure, ensuring that the model's pre-existing reasoning abilities are not compromised, which is well-suited as the final stage of post-training.

**Performance with different token budget.** We vary the generation-length budget during inference from short to long, and compare the change in accuracy and actual reasoning length of S-GRPO and vanilla CoT on GSM8K (representing simple problems) and AIME 2024 (representing complex problems). The results in Figure 3 show that, across all budgets tested, S-GRPO consistently achieves higher accuracy while generating shorter sequences compared to vanilla CoT, further highlighting the effectiveness of our method.

Moreover, we observe that under tight length budgets, S-GRPO achieves significantly higher accuracy than vanilla CoT while generating sequences with comparable length. Differently, under larger length budgets, S-GRPO produces significantly shorter reasoning paths with slightly better accuracy, compared with vanilla CoT. The trends in both accuracy and actual generation length of S-GRPO are smoother than vanilla CoT, indicating its greater robustness to variations in length budget. Overall, S-GRPO achieves high accuracy under a low length budget, suggesting that our method is capable of generating concise yet accurate reasoning paths.

**Ablation results.** To verify the effectiveness of each design component in S-GRPO, we conduct ablation studies under three different settings. *w/o. Decaying (Shortest 1)* denotes the setting where only the shortest correct response in the serial group is assigned a reward of 1, while all other responses receive a reward of 0. *w/o. Decaying (All 1)* refers to the configuration where all correct responses in the serial group are assigned a reward of 1, while incorrect ones receive a reward of 0. *w/o. Serial* indicates that we further remove the Serial-Group Generation mechanism based on the w/o. Decaying (All 1) configuration

Table 2: Ablation results on Qwen3-8B.

| Method | GSM8K | | AIME 2024 | | AMC 2023 | | MATH-500 | | GPQA | | Overall | |
|---|---|---|---|---|---|---|---|---|---|---|---|---|
| | Acc | Tokens | Acc | Tokens | Acc | Tokens | Acc | Tokens | Acc | Tokens | Acc | Tokens |
| *Qwen3-8B* | | | | | | | | | | | | |
| *S-GRPO* | 96.1 | 1,292 | 77.3 | 8,810 | 95.0 | 5,962 | 95.2 | 3,166 | 57.7 | 5,271 | **84.26** | 4,900 |
| *w/o. Decaying (Shortest 1)* | 95.9 | 1,175 | 69.6 | 8,721 | 92.5 | 4,581 | 94.8 | 2,740 | 55.7 | 4,734 | 81.70$_{-2.56}$ | **4,390**$_{-10.4\%}$ |
| *w/o. Decaying (All 1)* | 96.0 | 2,385 | 74.4 | 14,940 | 94.7 | 9,000 | 95.0 | 5,614 | 54.9 | 8,955 | 83.00$_{-1.26}$ | 8,179$_{+66.9\%}$ |
| *- w/o. Serial* | 95.8 | 2,355 | 72.7 | 15,154 | 92.8 | 8,983 | 94.4 | 5,440 | 55.8 | 8,819 | 82.30$_{-1.96}$ | 8,150$_{+66.3\%}$ |

The results in Table 2 indicate that rewarding only the shortest correct response imposes an overly strict constraint. Although this setting leads to further reductions in reasoning length, it comes at the cost of accuracy. And removing the design that assigns higher rewards to shorter outputs, i.e., *w/o. Decaying (All 1)*, results in lengthy reasoning. This is because long correct answers also receive high rewards, and the model does not shift in favor of generating short CoTs. When the Serial-Group Generation mechanism is removed, our method degenerates to GRPO, achieving performance comparable to *w/o. Decaying (All 1)* in both accuracy and reasoning length. This demonstrates that Serial-Group Generation, as an essential component of S-GRPO, does not compromise the model's exploration capability in reinforcement learning.

**Case study.** Figure 5 in the Appendix visually illustrates how S-GRPO generates answers at different early-exit positions and also shows the decaying reward assignments. For the first early-exit position, where the model produces an incorrect intermediate answer, we set the reward to 0. For subsequent exits that yield correct answers, we apply a decaying positive reward scheme, where earlier exits are associated with higher reward values. This design incentivizes the model to discover reasoning paths that are both accurate and succinct.

To more intuitively illustrate the effectiveness of our approach, a representative example is shown in Figure 4. The left and right sides of the figure compare the reasoning processes of vanilla CoT and S-GRPO. Although both methods yield the correct final answer, S-GRPO achieves this using less than half of the reasoning budget, showcasing its effectiveness in mitigating the overthinking problem [11]. The central portion of the figure displays an early exit obtained by directly truncating the vanilla CoT's reasoning process using the same token budget as S-GRPO. The reasoning model fails to reach the correct conclusion based on the available partial reasoning. This demonstrates that S-GRPO effectively identifies the correct solution path and guides the model to concise and accurate reasoning, while inherently avoiding the underthinking issue of superficial exploration [30].

## 4 Related Work

The success of OpenAI o1 [4, 31] highlights the powerful potential of reinforcement learning in post-training to enhance model reasoning capabilities. With the open source of Deepseek-R1 [3] and Qwen3 [21], reasoning models are now widely deployed locally, drawing the LLM community's attention to the efficiency of long chain-of-thought generation.

**Reinforcement learning for reasoning.** Rule-based outcome reward RL [3, 32, 33, 34, 35, 36, 37] has emerged as the mainstream approach for post-training optimization of reasoning models. This paradigm simplifies reward design by employing binary 0/1 rewards determined through rule-based correctness evaluation, eliminating the need for separate reward models as required in original GRPO [6, 3, 38] implementations, thereby substantially reducing memory and computational overhead during RL training. Recent advances have introduced numerous RL algorithm variants focusing on training efficiency [39, 40, 41], value function optimization [42, 43, 44], and other aspects. However, these developments preserve GRPO's fundamental mechanism that samples several CoTs in parallel for a group, consequently overlooking optimization opportunities during sequential thought generation within a single CoT. Our proposed optimized version of GRPO, S-GRPO, maintains the rule-based outcome reward framework while enabling intermediate reasoning process rewards by sampling multiple early-exit thought chains in serial for one CoT. We allow existing reasoning models to effectively mitigate overthinking and boost inference efficiency.

**Efficient reasoning.** Approaches for efficient reasoning can be categorized into two groups: training-free and training-based methods. Training-free methods typically improve reasoning effi-

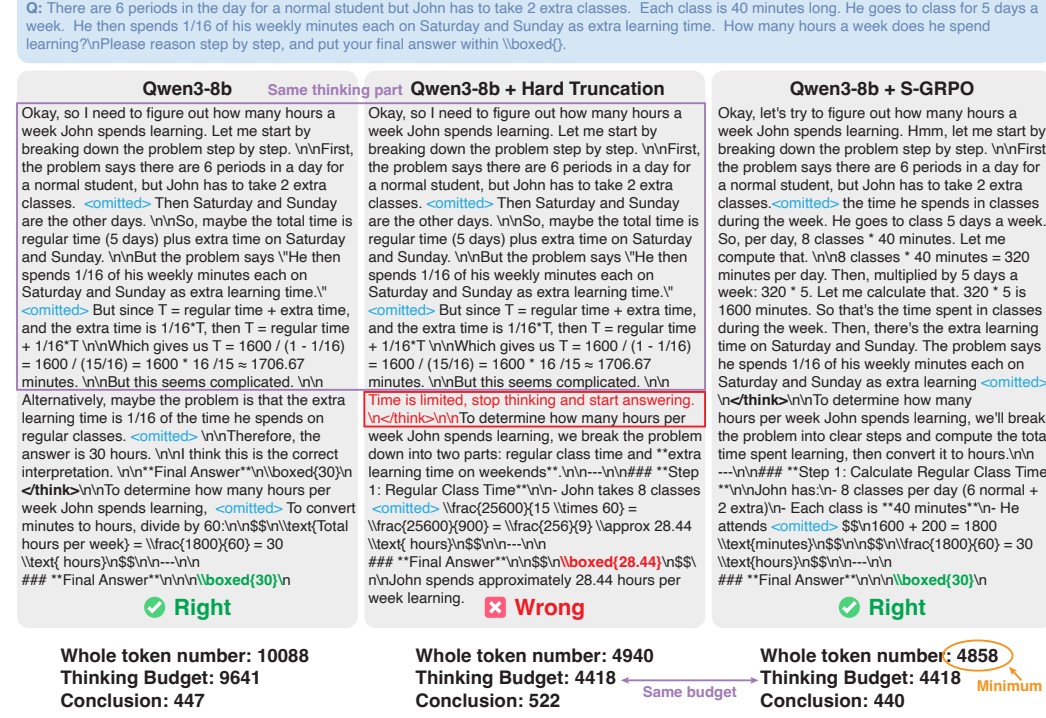

Figure 4: Comparison of a generated content sample on GSM8K.

ciency through dynamic prompting strategies [45, 46, 47, 48, 49, 50, 51, 52] , Best-of-N sampling pruning [53, 54] and optimizations [55, 56, 57] , and early-exit [58, 15] mechanisms during reasoning. DEER [15] and our method employ a similar operation by implementing early exit during CoT generation and producing intermediate answers. The key distinction lies in that DEER directly makes early-exit decisions during inference based on the confidence of intermediate answers, whereas our method leverages RL to reinforce models' behavior of correct early exit, which requires no additional compute overhead during inference.

Training-based methods primarily supervised fine-tuning models with variable-length CoT data [59, 60, 61, 62, 63, 64, 45], or training with length-rewards RL [12, 65, 66, 67, 68, 69, 70, 71, 72]. Recently, ConCISE [26] constructs concise CoT data by inserting prompt tokens and employing early-exit during inference, then enhances the model's reasoning conciseness through SFT/SimPO [73, 74]. [27] and [28] upweight rewards for the shorter CoTs, having correct answers, in parallel-sampled reasoning chains. However, they inherit GRPO's parallel sampling paradigm that neglects the reward function's attention to the intermediate serial reasoning processes.

## 5   Conclusion

This paper introduces Serial-Group Decaying-Reward Policy Optimization (S-GRPO), which innovatively leverages rule-based outcome rewards to regulate intermediate reasoning processes. By incentivizing the LLM to produce high-quality thoughts earlier and exit promptly when the generated thought is sufficient, S-GRPO enhances reasoning efficiency and maintains accuracy. Empirical evaluations demonstrate that S-GRPO achieves optimal synergy between efficiency and accuracy, significantly outperforming existing efficient reasoning methods. Specifically, S-GRPO demonstrates compatibility with state-of-the-art reasoning models, including Qwen3 and Deepseek-distill, achieving a 40.4%~61.1% reduction in sequence length while improving accuracy (absolute) by 0.72%~3.92% across five benchmark datasets: GSM8K, AIME 2024, AMC 2023, MATH-500, and GPQA Diamond. S-GRPO is well-suited for deployment as the final optimization stage in post-training pipelines, offering a practical solution for enhancing reasoning efficiency.

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

# A   An example of Training Data

Here is an example that shows the training data truncation and decaying reward assignment.

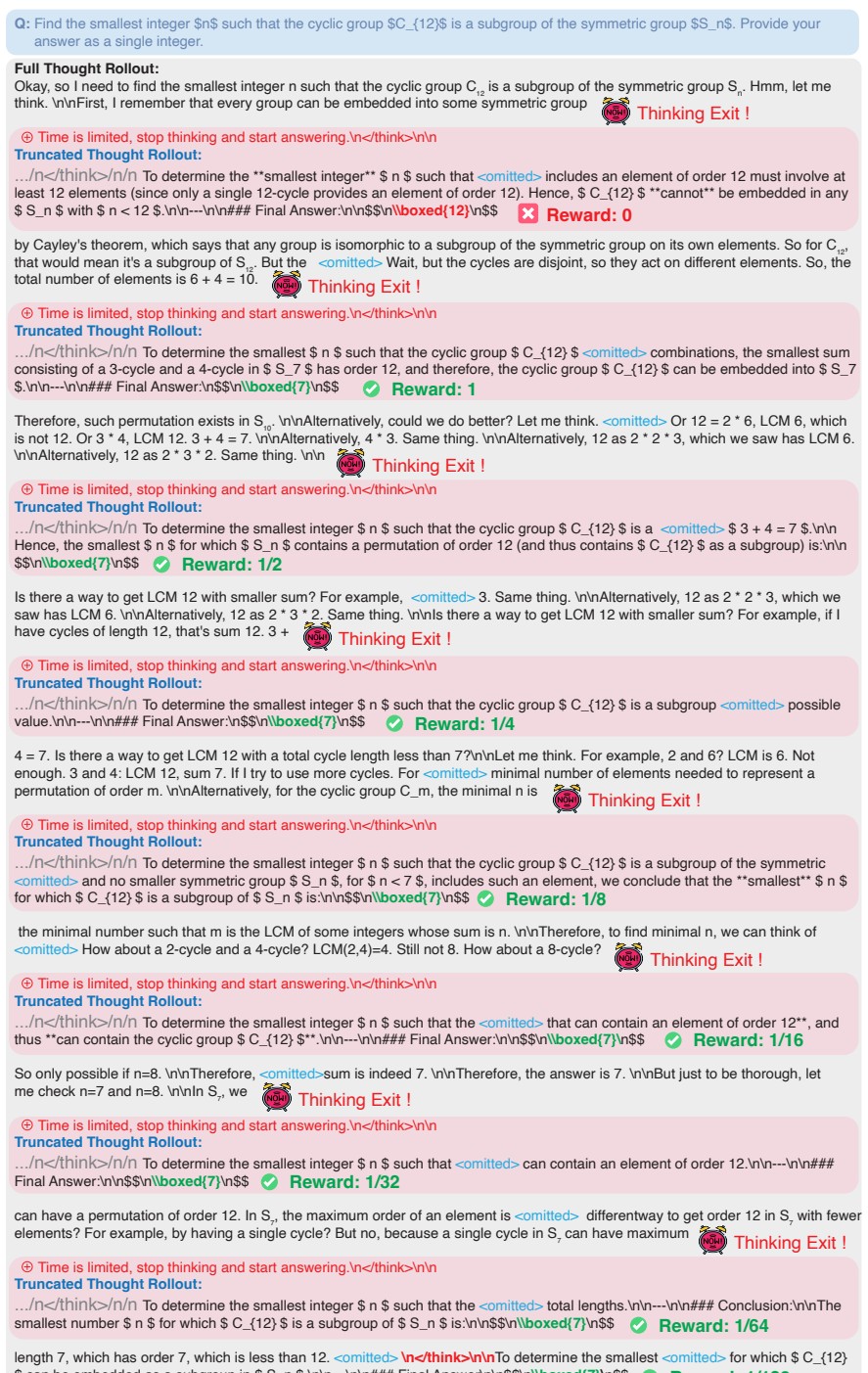

Figure 5: An example shows the training data truncation and decaying reward assignment.

## B    Computation Resource

In our experiments, $64 \times 80$g memory was used to train the models.

## C    Robustness to Prompt Template Variations

We evaluated the sensitivity of our thinking path truncation mechanism to different prompt templates. Three variations were tested: (1) an explicit time constraint prompt, (2) a detailed explanation prompt, and (3) a minimal delimiter-only prompt. All variants included the `\n</think>\n\n` delimiter.

Table 3: Performance across different prompt templates for thinking path truncation.

| Prompt Template | AIME 2024 | | MATH-500 | |
|---|---|---|---|---|
| | Acc | Tokens | Acc | Tokens |
| *Qwen3-8B with S-GRPO* | | | | |
| *Time is limited, stop thinking and start answering.* `\n</think>\n\n` | 77.3 | 8,810 | 95.2 | 3,166 |
| *Considering the limited time by the user, I have to give the solution based on the thinking directly now.* `\n</think>\n\n` | 77.1 | 8,780 | 95.0 | 3,202 |
| `\n</think>\n\n` | 76.9 | 8,823 | 95.3 | 3,154 |

The results demonstrate remarkable consistency across all prompt variations. Accuracy differences are negligible, and token usage remains stable. This indicates that the `\n</think>\n\n` delimiter, rather than the specific prompt wording, is the key factor in controlling thinking path truncation. Such robustness to prompt engineering choices enhances the practical applicability of our approach.

## D    Comparison of Truncation Strategies

We investigated different strategies for truncating thinking paths beyond random truncation. Specifically, we explored semantic-based truncation at specific markers that indicate thought shifts, such as "Wait" and "Alternatively" tokens, following similar approaches in prior work [75].

Table 4: Performance comparison of different truncation strategies.

| Truncation Strategy | AIME 2024 | | MATH-500 | |
|---|---|---|---|---|
| | Acc | Tokens | Acc | Tokens |
| *Qwen3-8B with S-GRPO* | | | | |
| *Random selection* | 77.3 | 8,810 | 95.2 | 3,166 |
| *"Wait" or "Alternatively" position* | 76.7 | 8,930 | 95.2 | 3,378 |

While semantic-based truncation achieved comparable performance to random truncation, it exhibited critical stability issues during training. As the model improves through training iterations, reasoning chains become increasingly compressed and streamlined, resulting in a significant reduction of transition tokens like "Wait" and "Alternatively". This progressive decrease creates a practical limitation: for certain queries, insufficient semantic markers are available to meet the sampling requirements for the second rollout phase (requiring $n$ samples). Consequently, semantic-based truncation becomes unreliable as training progresses.

In contrast, random truncation maintains consistent applicability throughout training regardless of how compressed the reasoning becomes. The combination of stable training dynamics and comparable performance validates our choice of random truncation as the primary strategy in the implementation of S-GRPO.

## E    Computational Efficiency of Dual-Rollout Training

We measured the actual time cost of S-GRPO's dual-rollout approach compared to standard GRPO on Qwen3-8B training with rollout n=8.

Despite requiring two inference passes, S-GRPO incurs only 20% additional time overhead rather than the 2× increase. This efficiency comes from two factors: (1) The second rollout only generates the

Table 5: Time cost comparison between GRPO and S-GRPO.

| Method | Average Rollout Time (s) | Average Time Increase | Relative Overhead |
|---|---|---|---|
| *Qwen3-8B Training* | | | |
| *GRPO (single rollout)* | 457 | - | 1.00× |
| *S-GRPO (dual rollout)* | 550 | +93s | 1.20× |

shorter conclusion portion, not the full response, reducing decoding time proportionally to sequence length; (2) The truncated thinking path uses efficient prefill computation rather than sequential decoding.

## F    Effectiveness Across Different Domains

Beyond scientific and mathematical reasoning, we extended our evaluation to MMLU-Pro (general knowledge) and LiveCodeBench v5 (code generation, 2024.10-2025.02).

Table 6: Performance of S-GRPO across different task domains.

| Model | MMLU-Pro | | LiveCodeBench v5 | |
|---|---|---|---|---|
| | Acc | Tokens | Acc | Tokens |
| *DeepSeek-R1-Distill-Qwen-7B* | | | | |
| *Vanilla* | 52.4 | 3,646 | 35.8 | 12,255 |
| *S-GRPO* | 52.4 | 1,597 | 35.9 | 7,692 |
| *Qwen3-8B* | | | | |
| *Vanilla* | 75.1 | 4,422 | 56.2 | 14,794 |
| *S-GRPO* | 75.2 | 2,313 | 56.2 | 10,203 |

The results demonstrate that S-GRPO also maintains accuracy while achieving substantial token reduction across other domains. On MMLU-Pro, token usage decreased by 56.2% and 47.7% for the two models respectively, with accuracy preserved or slightly improved. Similarly, on LiveCodeBench, S-GRPO reduced tokens by 37.2% and 31.0% while maintaining code generation performance.

## G    Limitations and Future works

In the later stages of training, S-GRPO encounters a bottleneck: the average reward begins to decline as the response length shortens. This is because the randomly truncated early stopping points become increasingly short, and excessively short CoTs are less likely to generate correct answers. In future work, optimizing the truncation strategy to replace the uniform sampling of truncation points could provide improvements.

## H    Response Length and Accuracy Curve

The Figure 6 below illustrates the response length and accuracy trends on the AIME 2024 dataset across different training steps for models trained with the RL with length penalty method and the S-GRPO method. The base model used is DeepSeek-R1-Distill-Qwen-7B. Both methods demonstrate a similar downward trend in response length. However, the RL with length penalty method exhibits an earlier decline in accuracy, highlighting the superior performance of the S-GRPO approach.

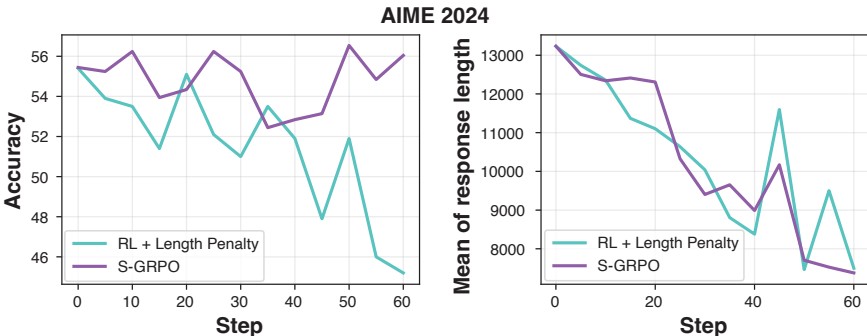

Figure 6: The response length and accuracy trends on the AIME 2024 dataset across different training steps for models trained with the RL with length penalty method and the S-GRPO method.

# I  More Case Study

To further demonstrate the effectiveness of S-GRPO in comparison to vanilla CoT and its truncated variant, we present two additional examples from the AIME 2024 benchmark in Figure 7. These cases highlight the advantages of S-GRPO in solving complex reasoning tasks with a significantly reduced token budget while maintaining accuracy.

In the first example (top row), all three methods attempt to find the greatest real number satisfying a geometric constraint. The vanilla CoT model produces the correct answer but requires a substantial thinking budget (12,015 tokens). When truncated to match the thinking budget of S-GRPO (6,167 tokens), the vanilla CoT fails to reach the correct conclusion. In contrast, S-GRPO successfully identifies the correct solution path and arrives at the correct answer using only 6,167 thinking budget (7,214 total token budget), demonstrating its ability to achieve efficiency without sacrificing accuracy.

In the second example (bottom row), the task involves calculating the length of a geometric segment based on given constraints. Vanilla CoT again fails when truncated to the same thinking budget as S-GRPO (9,333 tokens), producing an incorrect answer. Meanwhile, S-GRPO reaches the correct solution with a minimal total token budget (10,836 tokens), significantly outperforming vanilla CoT's full reasoning process, which consumes 28,171 tokens without arriving at the correct solution.

These examples further validate that S-GRPO effectively balances reasoning depth and efficiency, avoiding overthinking. By guiding the model to concise and accurate reasoning paths, S-GRPO consistently outperforms baseline methods in challenging reasoning tasks.

# J  More Related Work

**Dynamic and early-exit architectures.**  There are works exploring dynamic neural networks and early-exit mechanisms to improve inference efficiency. Dynamic neural networks [76] provide a general framework for adaptively adjusting computation based on input complexity, enabling selective activation of subnetworks or layers during inference. Building upon this concept, LayerSkip [77] introduces inter-layer early-exit strategies that allow large language models to skip redundant transformer layers dynamically, achieving faster decoding without substantial performance loss. While LayerSkip focuses on inter-layer early-exit, S-GRPO operates at the reasoning-path level.

More recently, hybrid approaches such as *A Stitch in Time Saves Nine* [78] leverage small vision–language models to guide the early termination of reasoning in larger models, serving as precise external controllers that accelerate computation. These studies collectively demonstrate the potential of adaptive computation and early-exit mechanisms to balance efficiency and accuracy in large-scale models, inspiring further exploration into reasoning-level early stopping and adaptive generation. S-GRPO eliminates the need for small model assistance by internalizing early-exit capabilities directly into the reasoning model through reinforcement learning.

**Q:** Let $A$, $B$, $C$, and $D$ be point on the hyperbola $\frac{x^2}{20} - \frac{y^2}{24} = 1$ such that $ABCD$ is a rhombus whose diagonals intersect at the origin. Find the greatest real number that is less than $BD^2$ for all such rhombi.\nPlease reason step by step, and put your final answer within \boxed{}.

| Qwen3-8b | Same thinking part Qwen3-8b + Hard Truncation | Qwen3-8b + S-GRPO |
|---|---|---|
| Okay, so I need to find the greatest real number that is less than BD² for all such rhombi ABCD on the hyperbola x²/20 - y²/24 = 1. The rhombus has diagonals intersecting at the origin. Let me try to visualize this first. \n\nA rhombus has all sides equal, and its <omitted> Which is the same condition as before. So, ac + bd = 0. Therefore, this condition is equivalent to diagonals being perpendicular. Therefore, the condition that ABCD is a rhombus is equivalent to ac + bd = 0, given <omitted> Therefore, the domain of k is sqrt(5/6) < lkl < sqrt(6/5). Therefore, k is in (-sqrt(6/5), -sqrt(5/6)) ∪ (sqrt(5/6), sqrt(6/5)).\n\nBut since S(k) is even in k (since replacing k by -k doesn't change S(k)), we can consider k > 0, so k ∈ (sqrt(5/6), sqrt(6/5)).\n\nTherefore, we need to minimize S(k) = 120(k² + 1)/(6k² - 5) for k ∈ (sqrt(5/6), sqrt(6/5)).\n\nLet me denote t = k². Then, since k ∈ (sqrt(5/6), sqrt(6/5)), then t ∈ (5/6, 6/5). Therefore, S(t) = 120(t + 1)/(6t - 5). <omitted> that is less than BD² for all such rhombi. Therefore, the answer is \boxed{480}.\n\n**Final Answer**\n\boxed{480}\n</think>\n\nTo solve the problem, we are given a hyperbola:\n\n$$\frac{x^2}{20} - \frac{y^2}{24}$ <omitted> real number less than ****$BD^2$**** for all such rhombi** is:\n\n\n**\boxed{480}**\n | Okay, so I need to find the greatest real number that is less than BD² for all such rhombi ABCD on the hyperbola x²/20 - y²/24 = 1. The rhombus has diagonals intersecting at the origin. Let me try to visualize this first. \n\n A rhombus has all sides equal, and its <omitted> Which is the same condition as before. So, ac + bd = 0. Therefore, this condition is equivalent to diagonals being perpendicular. Therefore, the condition that ABCD is a rhombus is equivalent to ac + bd = 0, given <omitted> Therefore, the domain of k is sqrt(5/6) < lkl < sqrt(6/5). Therefore, k is in (-sqrt(6/5), -sqrt (5/6)) ∪ (sqrt(5/6), sqrt(6/5)).\n\nBut since S(k) is even in k (since replacing k by -k doesn't change S(k)), we can consider k > 0, so k ∈ (sqrt(5/6), sqrt(6/5)).\n\nTherefore, we need to minimize S(k Time is limited, stop thinking and start answering.\n</think>\n\nTo solve the problem, we analyze the geometry of a rhombus $ABCD$ inscribed in the hyperbola $\frac{x^2}{20} - \frac{y^2}{24} = 1$, with diagonals intersecting at the = 264 \cdot \frac{5}{11} = 120\n$$\n\n---\n\n### Final Answer\n\nTherefore, the greatest real number less than $BD^2$ for all such rhombi is the **minimum value of $BD^2$**, which is:\n\n$$\n\boxed{120}\n$$ | Okay, so I need to find the greatest real number that is less than BD² for all rhombi ABCD on the hyperbola x²/20 - y²/24 = 1, where the diagonals intersect at the origin. Hmm, let me start by recalling some properties of rhombi and hyperbolas. <omitted> Therefore, given that OA is perpendicular to OB, and both A and B lie on the hyperbola. So, perhaps we can parametrize points A and B such that OA ⊥ OB.\n\n<omitted>\n\nBut let me check with specific values. Let me take k approaching 720/11. Let me compute 720 /11 ≈ 65.4545. Let me take k = 720/11 + ε, where ε approaches 0+. Then BD² approaches 480. Therefore, yes, 480 is the infimum. Therefore, the answer is 480. Therefore, I think the answer is \boxed{480}.\n\n**Final Answer**\n\boxed{480}\n</think>\n\nTo solve the problem, we are given a hyperbola:\n\n$$\frac{x^2}{20} - \frac{y^2}{24} = 1$$\n\nand a rhombus $ABCD$ inscribed in this hyperbola, with diagonals intersecting at the origin. We are to find the greatest <omitted> Since $BD^2$$ can get arbitrarily close to 480 but never actually reaches it (as $$c^2$$ must be strictly greater than $\frac{720}{11}$), the greatest real number less than $BD^2$$ for all such rhombi is:\n\n\n\boxed{480}\n |
| ✅ **Right** | ❌ **Wrong** | ✅ **Right** |
| **Whole token number: 13877**
**Thinking Budget: 12015**
**Conclusion: 1862** | **Whole token number: 7624**
**Thinking Budget: 6167** ← Same budget
**Conclusion: 1457** | **Whole token number: 7214** Minimum
**Thinking Budget: 6167**
**Conclusion: 1047** |

**Q:** Rectangles $ABCD$ and $EFGH$ are drawn such that $D,E,C,F$ are collinear. Also, $A,D,H,G$ all lie on a circle. If $BC=16$,$AB=107$,$FG=17$, and $EF=184$, what is the length of $CE$?\nPlease reason step by step, and put your final answer within \boxed{}.

| Qwen3-8b | Same thinking part Qwen3-8b + Hard Truncation | Qwen3-8b + S-GRPO |
|---|---|---|
| Okay, so I need to find the length of CE given these rectangles and some collinear points. Let me try to visualize this first. There are two rectangles, ABCD and EFGH. The points D, E, C, F are all on the same straight line. Also, points A, D, H, G lie on a circle. The sides BC = 16, AB = 107, FG = 17, and EF = 184. Hmm.\n\nFirst, let me sketch this mentally. <omitted> if E is at ≈ -184.09, then F is at ≈ -0.09, which is just left of D(0,0). So, the order would be E, F, D, C. So, D is between F and C. But the problem <omitted> solving gives x = [398 ± sqrt(398² - 4*31 120)] / 2 = [398 ± sqrt(158404 - 124480)] / 2 = [3 98 ± sqrt(33924)] / 2 = [398 ± 184.185]/2. Which gives two solutions:\n\n(398 + 184.185)/2 ≈ 582. 185 / 2 ≈ 291.09\n\n(398 - 184.185)/2 ≈ 213.815 / 2 ≈ 106.9075.\n\nSo, CE is either ≈ 291.09 or ≈ 106.91. But how to choose?\n\nWait, maybe there 's another condition. For example, <omitted> result. \n\n\n**Final Answer**\n\boxed{199 - \sqrt{8481}}\n</think>\n\nTo solve the problem, we are given two rectangles, $ABCD$ and $EFG$ <omitted> a circle.\n\nThus, the correct value is:\n\n$$\nCE = 199 - \sqrt{8481}\n$$\n\n---\n\n### Final Answer\n\n\n**\boxed{199 - \sqrt{8481}}**\n | Okay, so I need to find the length of CE given these rectangles and some collinear points. Let me try to visualize this first. There are two rectangles, ABCD and EFGH. The points D, E, C, F are all on the same straight line. Also, points A, D, H, G lie on a circle. The sides BC = 16, AB = 107, FG = 17, and EF = 184. Hmm.\n\nFirst, let me sketch this mentally. <omitted> if E is at ≈ -184.09, then F is at ≈ -0.09, which is just left of D(0,0). So, the order would be E, F, D, C. So, D is between F and C. But the problem <omitted> solving gives x = [398 ± sqrt(398² - 4*31 120)] / 2 = [398 ± sqrt(158404 - 124480)] / 2 = [3 98 ± sqrt(33924)] / 2 = [398 ± 184.185]/2. Which gives two solutions:\n\n(398 + 184.185)/2 ≈ 582. 185 / 2 ≈ 291.09\n\n(398 - 184.185)/2 ≈ 213.815 Time is limited, stop thinking and start answering.\n</think>\n\nTo solve the problem, we begin by analyzing the given geometric configuration involving two rectangles, $ABCD$ and $EFGH$, with the following known values:\n\n- $AB = 107$, $BC = 16$, so $ABCD$ <omitted> reasonable given the context, we choose:\n\n$$\n\boxed{ 199 - \sqrt{8481}}$$\n\n---\n\n### Final Answer:\n\n\n$$\n\boxed{199 - \sqrt{8481}}$$ | Okay, so I need to find the length of CE given these rectangles and some collinear points. Let me try to visualize this first. There are two rectangles, ABCD and EFGH. The points D, E, C, F are all on the same straight line. Also, points A, D, H, G lie on a circle. The sides BC = 16, AB = 107, FG = 17, and EF = 184. I need to find CE. \n\nFirst, let me sketch this mentally. Let me recall that rectangles have opposite sides equal and all <omitted> = 0. \n\nPlug in H(e, -17): e² + 289 + 2G e - 16*(-17) = 0 \n\nWhich is e² + 289 + 2G e + 272 = 0 \n\nTherefore, e² + 561 + 2G e = 0 => 2G e = -e² - 561 => G = (-e² - 561)/(2e) \n\n<omitted> is below, we have exact values, whereas in the case above, we had irrational numbers. CE = 104. \n\n**Final Answer**\n\boxed{104}\n</think>\n\nTo solve the problem, we are given two rectangles, $ABCD$ and $EFGH$, with the following properties:\n- $D$, E, C, F$$ are collinear.\n- $$A, D, H, G$$ lie on a circle.\n- Side <omitted> places point $$E$$$ between $$D$$ and $C$, and $$F$$$ beyond $C$, this is a more natural configuration for the given side lengths and collinearity.\n\nThus, the correct value is:\n\n\n\boxed{104}\n |
| ❌ **Wrong** | ❌ **Wrong** | ✅ **Right** |
| **Whole token number: 28171**
**Thinking Budget: 26555**
**Conclusion: 1616** | **Whole token number: 11239**
**Thinking Budget: 9333** ← Same budget
**Conclusion: 1906** | **Whole token number: 10836** Minimum
**Thinking Budget: 9333**
**Conclusion: 1503** |

Figure 7: Comparison of two generated content samples on AIME 2024.