# OpenReview forum: "S-GRPO: Early Exit via Reinforcement Learning in Reasoning Models"
_NeurIPS.cc/2025/Conference — NeurIPS 2025 poster_

### Official Review · Reviewer_N5Z5 · 2025-06-14

**Clarity:** 3
**Significance:** 3
**Originality:** 3
**Rating:** 5
**Confidence:** 4

**Summary:**

The authors propose two techniques to solve the long-answer problem in reasoning models: short responses generation and decaying reward assignment. The method enjoys shorter response length and better result.

**Questions:**

See Weaknesses.

**Ethical Concerns:**

["NO or VERY MINOR ethics concerns only"]

**Final Justification:**

The author's response addressed my concerns. I will keep my score.

**Limitations:**

Yes

**Quality:**

3

**Strengths And Weaknesses:**

Strengths:
1. The method is novel as far as I know. Inserting a termination sentence in the middle of the response is interesting.
2. The writing is crystal clear. I enjoy reading the paper.
3. The method is significant. Long response is a fatal problem for reasoning models.
4. The quality is good. The authors did solid ablation study to verify the contribution of each part. And the experimental result looks solid.

Weaknesses:
1. There is one more ablation study the authors may need to do. The whole process in the paper sounds generating the response twice: first getting the reasoning path and truncating it to insert the early termination sentence. This may have twice of the inference cost in training. More generations do have benefit in training.The authors should compare the results with double generation of the responses in training.
2. It's unclear whether the prompt in termination sentence matter "Time is limited, stop thinking and start answering." Do the authors try different templates? What if the authors add this sentence to the end of normal training of reasoning models and ask the model to generate again?
3. Math is a relatively easy task. What's the method's performance on code related tasks?

---

> ### Author Rebuttal · Authors · 2025-07-30
>
> Thanks for your suggestions.
> ## W1:
> Double inference time during training? & Comparison with double completions.
> ## Reply to W1:
> 1. We measure the actual time cost of GRPO and S-GRPO with "rollout n=8" on Qwen3-8B. It shows that GRPO's single rollout takes an average of approximately 457s, while S-GRPO's two rollouts take around 550s in total, representing only about a 20% increase in inference time, far below the intuitive estimate of "doubling". S-GRPO's two rollouts do not double inference time because:
> - The decoding time is proportional to the response length, as it is a length-bounded process. The second rollout only regenerates the conclusion portion, which is substantially shorter than generating the full thinking path + conclusion from scratch. Since decoding time scales with sequence length, this second phase incurs much lower computational cost.
> - The query + thinking path portion uses prefill (rather than decode), which is computationally efficient and doesn't significantly impact overall timing.
>
> 2. We conduct the suggested ablation by doubling the rollout number (n→2n) for the "RL + Length Penalty" baseline from Table 1 in the main text:
>
> | Model | Method | Rollout n | Generation Time | AIME 2024 Acc | MATH-500 Acc |
> |-------|--------|---|----------------|-----------|----------|
> | Qwen3-8B | RL + Length Penalty | 8 | 450s | 73.8 | 94.2 |
> | Qwen3-8B | RL + Length Penalty | 16 | 763s | 73.5 | 94.3 |
> | Qwen3-8B | S-GRPO | First rollout n = 1; Second rollout n = 8 | 550s | 77.3 | 95.2 |
>
> It shows that doubling n for "RL + Length Penalty" increases generation time by about 70%, but the performance remains similar, still failing to maintain accuracy with length degradation. This demonstrates that S-GRPO's two-phase generation is more efficient and effective than simple scaling of generation quantity.
>
> ## W2:
> Try different prompt templates. & Influence of adding "Time is limited...`\n</think>\n\n`" to the end of normal training of reasoning models.
>
> ## Reply to W2:
> We experiment with different prompt templates for truncating the thinking path. Our results show that as long as the `\n</think>\n\n` is present, the model terminates its reasoning and proceeds to generate a conclusion. Moreover, the specific wording of the prompt hardly affects the results, as shown in the table below.
>
> | Model | Method | Prompt | AIME 2024 Acc | AIME 2024 Tokens | MATH-500 Acc | MATH-500 Tokens |
> | ----- | ------ | ------ | ------------- | ---------------- | ------------ | --------------- |
> | Qwen3-8B | S-GRPO | Time is limited, stop thinking and start answering.`\n</think>\n\n` | 77.3 | 8,810 | 95.2 | 3,166 |
> | Qwen3-8B | S-GRPO | Considering the limited time by the user, I have to give the solution based on the thinking directly now.`\n</think>\n\n` | 77.1 | 8,780 | 95.0 | 3,202 |
> | Qwen3-8B | S-GRPO | `\n</think>\n\n` | 76.9 | 8,823 | 95.3 | 3,154 |
>
> For the question of “add this sentence to the end of normal training of reasoning models and ask the model to generate again”:
> 1. When using this way during inference,  it is conceptually similar to the think budget truncation method in "Qwen3 Technical Report". It could prompt the model to stop thinking and generate a conclusion, but often at the cost of degraded accuracy. As shown in Figure 4 of the main paper and Figure 7 of Appendix E, adding this prompt to a normally trained model can affect the performance, especially when the reasoning process is not explicitly trained to accommodate such abrupt termination.
> 2. When using this way during normal GRPO training, it is equivalent to "S-GRPO without reward decay", which is shown in the ablation study in the main text (Table 2, Row "w/o. Decaying (All 1)"). We also show the results in the table below. The results show that "w/o. Decaying (All 1)" results in lengthy reasoning, and our complete S-GRPO is better than "w/o. Decaying (All 1)" on all five benchmarks.
>
> | Method | GSM8K |  | AIME 2024 |  | AMC 2023 |  | MATH-500 |  | GPQA |  | Overall |  |
> |--------|-------|-------|-----------|-------|----------|-------|----------|-------|------|-------|---------|-------|
> |        | Acc | Tokens | Acc | Tokens | Acc | Tokens | Acc | Tokens | Acc | Tokens | Acc | Tokens |
> | Qwen3-8B (S-GRPO) | 96.1 | 1,292 | 77.3 | 8,810 | 95.0 | 5,962 | 95.2 | 3,166 | 57.7 | 5,271 | 84.26 | 4,922 |
> | Qwen3-8B (S-GRPO w/o. Decaying (All 1)) | 96.0 | 2,385 | 74.4 | 14,940 | 94.7 | 9,000 | 95.0 | 5,614 | 54.9 | 8,955 | 83.00 | 8,179 |
>
> ## W3:
> Math is a relatively easy task. What's the method's performance on code-related tasks?
> ## Reply to W3:
> Besides math tasks, we have evaluated science tasks such as the GPQA benchmark in the main text. And we have further evaluated two models, DeepSeek-R1-Distill-Qwen-7B and Qwen3-8B, on both general knowledge tasks (MMLU-Pro) and code-related tasks (LiveCodeBench v5, 2024.10-2025.02), as shown in the table below.
>
> | Model                  | MMLU-Pro Acc | MMLU-Pro Tokens | LiveCodeBench (v5, 2024.10-2025.02) Acc | LiveCodeBench (v5, 2024.10-2025.02) Tokens |
> | ---------------------- | -------- | ----------- | ------- | ---------- |
> | **DeepSeek-R1-Distill-Qwen-7B (Vanilla)**       | 52.4    | 3,646       | 35.8   | 12,255     |
> | **DeepSeek-R1-Distill-Qwen-7B (S-GRPO)** | 52.4    | 1,597       | 35.9   | 7,692      |
> | **Qwen3-8B (Vanilla)** | 75.1    | 4,422       | 56.2   | 14,794     |
> | **Qwen3-8B-SGRPO (S-GRPO)**     | 75.2    | 2,313       | 56.2   | 10,203     |
>
>
> Our results show that on both the general knowledge task (MMLU-Pro) and the code-related task (LiveCodeBench v5, 2024.10-2025.02), S-GRPO significantly reduces output length while maintaining comparable accuracy, demonstrating effective compression.

---

> > ### Comment · Reviewer_N5Z5 · 2025-08-04
> > **Thanks for the response**
> >
> > Thanks for the response. I have no further questions and I will keep my score.

---

> > > ### Author Response · Authors · 2025-08-07
> > > **Thank you for your review**
> > >
> > > We are delighted that our responses have successfully addressed all your concerns. We greatly appreciate your constructive comments, and thank you again for your positive comments.

---

### Official Review · Reviewer_Rkde · 2025-06-28

**Clarity:** 4
**Significance:** 4
**Originality:** 4
**Rating:** 5
**Confidence:** 4

**Summary:**

The paper introduces Serial-Group Decaying-Reward Policy Optimization (S-GRPO), a new RL training paradigm aimed at addressing the overthinking problem in current reasoning models during inference.

Innovations of the Paper

- Serial-Group Generation: Unlike GRPO, which parallelly samples multiple reasoning paths, S-GRPO forms a complete reasoning path. It then truncates this path at different positions to generate a series of intermediate results, constituting a "serial group."

- Decaying Reward Strategy: Rewards are assigned to the answers in the serial group based on their truncation positions, encouraging the model to exit as early as possible.

Experimental results show that when combined with models such as Qwen3 and Deepseek-distill, S-GRPO achieves a sequence length reduction of 35.4% to 61.1% while increasing accuracy by 0.72% to 6.08% on benchmarks including GSM8K, AIME 2024, AMC 2023, MATH-500, and GPQA.

**Questions:**

none

**Ethical Concerns:**

["NO or VERY MINOR ethics concerns only"]

**Final Justification:**

The author's response has successfully resolved my concerns. I will keep my score and recommend for acceptance.

**Limitations:**

yes

**Quality:**

4

**Strengths And Weaknesses:**

Strengths

- Innovative Serial-Group and Decaying Reward Mechanism: By serializing intermediate reasoning steps and using decaying rewards, the model is guided to exit early when reasoning is sufficient.

- Improved Efficiency and Accuracy: It significantly reduces reasoning length while enhancing accuracy, outperforming baseline methods (e.g., DEER, GRPO), with notable advantages in complex reasoning tasks (e.g, AIME 2024).

- Adequate Experiments: Effectiveness is validated across multiple mainstream reasoning models (e.g, Qwen3, Deepseek-distill) and in both mathematical and scientific reasoning tasks.

Weaknesses

- Task Domain Limitation: Validation experiments are primarily focused on mathematical and scientific reasoning tasks; effectiveness in other domains remains untested.

- Complex Hyperparameter Tuning: The reward function formula and sampling strategy for truncation positions are relatively complex, requiring separate tuning for different tasks.

---

> ### Author Rebuttal · Authors · 2025-07-30
>
> Thanks for your suggestions.
> ## W1:
> Effectiveness in other domains.
> ## Reply to W1:
> Following the works of [1] and [2], most of the optimization work of the R1-like model uses math and science as the main benchmark. Considering the space of paper, the evaluation of more scenarios was not shown in the main text. We have further evaluated two models, DeepSeek-R1-Distill-Qwen-7B and Qwen3-8B, on both general knowledge tasks (MMLU-Pro) and code-related tasks (LiveCodeBench v5, 2024.10-2025.02), as shown in the table below.
>
> | Model                  | MMLU-Pro Acc | MMLU-Pro Tokens | LiveCodeBench (v5, 2024.10-2025.02) Acc | LiveCodeBench (v5, 2024.10-2025.02) Tokens |
> | ---------------------- | -------- | ----------- | ------- | ---------- |
> | **DeepSeek-R1-Distill-Qwen-7B (Vanilla)**       | 52.4    | 3,646       | 35.8   | 12,255     |
> | **DeepSeek-R1-Distill-Qwen-7B (S-GRPO)** | 52.4    | 1,597       | 35.9   | 7,692      |
> | **Qwen3-8B (Vanilla)** | 75.1    | 4,422       | 56.2   | 14,794     |
> | **Qwen3-8B-SGRPO (S-GRPO)**     | 75.2    | 2,313       | 56.2   | 10,203     |
>
> Our results show that on both the general knowledge task (MMLU-Pro) and the code-related task (LiveCodeBench v5, 2024.10-2025.02), S-GRPO significantly reduces output length while maintaining comparable accuracy, demonstrating effective compression.
>
> ## W2:
> Need complex hyperparameter tuning?
> ## Reply to W2:
> 1. Compared with GRPO, our S-GRPO introduces minimal modifications in terms of hyperparameters. The main change lies in shifting from the traditional one rollout n setting to a two rollout n setting. In practice, we recommend fixing the first rollout number to 1 (focusing solely on the second rollout n), as the training goal of S-GRPO is to compress the thinking content. We experiment with different second rollout n, as shown in the table below. The results show that varying this hyperparameter (e.g., 8 or 16) yields stable performance across benchmarks, indicating that S-GRPO is not sensitive to this hyperparameter and does not require fine-grained tuning.
>
> | Model | Second rollout n | AIME 2024 Acc | AIME 2024 Tokens | MATH-500 Acc | MATH-500 Tokens |
> | ----- | ------ | ------------- | ---------------- | ------------ | --------------- |
> | Qwen3-8B | 8 | 77.3 | 8,810 | 95.2 | 3,166 |
> | Qwen3-8B | 16 | 76.7 | 8,730 | 95.4 | 3,198 |
>
> 2. S-GRPO inherits the core structure of GRPO and introduces only a lightweight extension through a decay-based reward function along the reasoning path. This design is general as the original GRPO and does not require task-specific tuning, making it applicable to a wide range of reasoning compression tasks.
>
> &nbsp;
> &nbsp;
>
> [1] ConCISE: Confidence-guided Compression in Step-by-step Efficient Reasoning
>
> [2] Training language models to reason efficiently

---

> > ### Comment · Reviewer_Rkde · 2025-08-05
> > **Response to authors**
> >
> > Thanks for the clarification. I will maintain my score and recommend for acceptance.

---

> > > ### Author Response · Authors · 2025-08-07
> > > **Thank you for your review**
> > >
> > > We are delighted that our responses have successfully addressed all your concerns. We greatly appreciate your constructive comments, and thank you again for your positive comments.

---

### Official Review · Reviewer_65Nr · 2025-06-30

**Clarity:** 3
**Significance:** 3
**Originality:** 3
**Rating:** 4
**Confidence:** 4

**Summary:**

The paper aims to implement an RL algorithm to make the LLM conduct reasoning efficiently. Based on the original group relative policy optimization (GRPO) algorithm, this paper proposes to form the trajectory group along the sequence length dimension, which only samples one reasoning path and serially selects multiple temporal positions from the path to exit thinking and directly generate answers. For correct answers within a serial group, rewards are gradually decreased when the sequence becomes longer. With S-GRPO, the reasoning models exhibit competitive reasoning performance while being much more efficient.

**Questions:**

From Figure 4, it seems that the model trained after S-GRPO will not output sentence like 'Time is limited, stop thinking and start answering.\n</think>\n\nTo' why does this not happen if the model is trained with every correct trajectories that ended with this sentence during thinking?

**Ethical Concerns:**

["NO or VERY MINOR ethics concerns only"]

**Final Justification:**

The author's response has resolved my concerns; according to other review comments and their corresponding responses, I tend to keep my score.

**Quality:**

3

**Strengths And Weaknesses:**

## Strengths

1. The method is simple yet effective.  The method simply conducts multiple early exits along one reasoning path, and uses exponential decay reward for correct answers to make it work. The experimental results show that the accuracies do not drop significantly with reasoning tokens are significantly decreased.
2. The experimental

## Weaknesses
1. Inefficiency in implementation. For each query, S-GRPO uses the model to infer one response first, then conduct early exit from different parts of the trajectory. Intuitively, this will roughly double the inference time at each iteration step during training.
2. The ablation study is a bit too simple. Intuitively, simply adding "Time is limited, stop thinking and start answering.\n</think>\n\n" would make the model learns to output this sentence at each trajectory, and the truncation is conducted randomly, is there any more elegant strategy that worth study?

---

> ### Author Rebuttal · Authors · 2025-07-30
>
> Thanks for your suggestions.
> ## W1:
> Double inference time during training？
> ## Reply to W1:
> We measure the actual time cost of GRPO and S-GRPO with "rollout n=8" on Qwen3-8B. It shows that GRPO's single rollout takes an average of approximately 457s, while S-GRPO's two rollouts take around 550s in total, representing only about a 20% increase in inference time, far below the intuitive estimate of "doubling".
> S-GRPO's two rollouts do not double inference time because:
> - The decoding time is proportional to the response length, as it is a length-bounded process. The second rollout only regenerates the conclusion portion, which is substantially shorter than generating the full thinking path + conclusion from scratch. Since decoding time scales with sequence length, this second phase incurs much lower computational cost.
> - The query + thinking path portion uses prefill (rather than decode), which is computationally efficient and doesn't significantly impact overall timing.
>
> ## W2:
> Influence of different prompt templates. & More discussion on truncation ways.
>
> ## Reply to W2:
> 1. We experiment with different prompt templates for truncating the thinking path. Our results show that as long as the `\n</think>\n\n` is present, the model terminates its reasoning and proceeds to generate a conclusion. Moreover, the specific wording of the prompt hardly affects the results, as shown in the table below.
>
> | Model | Method | Prompt | AIME 2024 Acc | AIME 2024 Tokens | MATH-500 Acc | MATH-500 Tokens |
> | ----- | ------ | ------ | ------------- | ---------------- | ------------ | --------------- |
> | Qwen3-8B | S-GRPO | Time is limited, stop thinking and start answering.`\n</think>\n\n` | 77.3 | 8,810 | 95.2 | 3,166 |
> | Qwen3-8B | S-GRPO | Considering the limited time by the user, I have to give the solution based on the thinking directly now.`\n</think>\n\n` | 77.1 | 8,780 | 95.0 | 3,202 |
> | Qwen3-8B | S-GRPO | `\n</think>\n\n` | 76.9 | 8,823 | 95.3 | 3,154 |
>
> 2. Beyond random truncation, we have also explored truncation at specific semantic markers, such as tokens indicating thought shifts like "Wait" and "Alternatively" [1]. As shown in the table below, these semantic-based truncation strategies achieve similar final performance to random truncation. However, the strategy of truncation at specific semantic markers is less stable during training: As training progresses, the reasoning chains become more compressed and streamlined, leading to a significant reduction in transition tokens like "Wait" and "Alternatively". This creates a practical limitation where, for certain queries, there are insufficient semantic markers to meet the sampling requirements for the second rollout phase (n samples), making this approach less stable for training. The stability advantage of random truncation, combined with its comparable performance, supports our choice of this strategy in the final S-GRPO implementation.
>
> | Model | Truncation position | AIME 2024 Acc | AIME 2024 Tokens | MATH-500 Acc | MATH-500 Tokens |
> | ----- | ------ | ------------- | ---------------- | ------------ | --------------- |
> | Qwen3-8B | Random selection | 77.3 | 8,810 | 95.2 | 3,166 |
> | Qwen3-8B | "Wait" or "Alternatively" position | 76.7 | 8,930 | 95.2 | 3,378 |
>
> ## Q1:
> Why are the specific prompts used during training (e.g., "Time is limited...`\n</think>\n\n`") not generated during inference?
> ## Reply to Q1:
> 1. One possible explanation for this phenomenon lies in the nature of reinforcement learning. We have experimented with training using truncated data via supervised fine-tuning (SFT), and under SFT, the model does learn to generate the sentence "Time is limited, stop thinking and start answering." However, this behavior does not emerge under reinforcement learning (RL). During RL training, both positive and negative samples (regarding adv > 0 is positive, adv < 0 is negative) are present. The sentence "Time is limited, stop thinking and start answering." may appear in both types of samples, leading to a gradient tradeoff that prevents the model from learning this pattern.
>
> 2. Another possible explanation is that S-GRPO is applied only in the final RL stage on a model that has already been extensively trained. This final RL stage is relatively short and primarily targets optimizing the model's overthinking behavior, rather than inducing significant shifts in its internal generation patterns.
>
> &nbsp;
> &nbsp;
>
> [1] s1: Simple test-time scaling

---

> > ### Comment · Reviewer_6wUr · 2025-08-07
> >
> > Thanks! I have carefully reviewed it.

---

> > > ### Author Response · Authors · 2025-08-07
> > > **Thank you for your review**
> > >
> > > We sincerely appreciate your thoughtful review and constructive feedback. We hope our responses have addressed your concerns to your satisfaction. If so, we would greatly appreciate your consideration in updating your evaluation. Should there be any remaining questions or points for clarification, we would be more than happy to continue the discussion.

---

### Official Review · Reviewer_6wUr · 2025-07-05

**Clarity:** 2
**Significance:** 3
**Originality:** 2
**Rating:** 4
**Confidence:** 3

**Summary:**

This paper proposes Serial-Group Decaying-Reward Policy Optimization (S-GRPO), a novel reinforcement learning paradigm that incentivizes reasoning models to exit early during chain-of-thought (CoT) generation, addressing the overthinking problem in large language models. By introducing a two-phase rollout procedure and a decaying reward strategy, S-GRPO improves reasoning efficiency and accuracy. Empirical results show significant accuracy gains (0.72%–6.08%) and sequence length reductions (35.4%–61.1%) across diverse benchmarks. The method is compatible with state-of-the-art reasoning models and offers a practical solution for enhancing reasoning efficiency in post-training pipelines.

**Questions:**

see weakness

**Ethical Concerns:**

["NO or VERY MINOR ethics concerns only"]

**Final Justification:**

After reviewing the rebuttal, I will be keeping my rating the same.

The authors have acknowledged the questions raised in the initial review. However, the rebuttal primarily points to existing results or promises future revisions to the paper rather than providing new evidence or experiments that would address the core concerns.

Specifically:

W1 (Length Penalties): The authors correctly point out that they have compared their method to length penalty approaches. While they argue their method is more nuanced, the fundamental concern about a simpler baseline remains.

W2 (Related Works): The commitment to expand the related works section is appreciated, but this does not change the substance of the paper as it stands.

W3 (Computational Resources): The clarification on GPU usage is helpful.

Overall, the rebuttal clarifies the authors' positions but does not substantially alter the assessment of the paper's contributions or its limitations. Therefore, my initial evaluation remains unchanged.

**Limitations:**

see weakness

**Quality:**

3

**Strengths And Weaknesses:**

Strengths：
Consistent accuracy improvements
Ablation studies and detailed comparisons with existing methods

Weaknesses:
Is it possible to directly apply a penalty for output length during the original GRPO training? Does the proposed method demonstrate superiority over this approach?


Early exiting is a common technique in dynamic neural networks and will be discussed in the related works.
For example,
[1]  Dynamic neural networks: A survey (small modles)
[2]LayerSkip: Enabling Early Exit Inference and Self-Speculative Decoding (language models)
[3] A Stitch in Time Saves Nine: Small VLM is a Precise Guidance for Accelerating Large VLMs (multi-model mdoels)

How many GPUs were used for training?

---

> ### Author Rebuttal · Authors · 2025-07-30
>
> Thanks for your suggestions.
> ## W1
> The performance of directly applying length penalties for RL.
> ## Reply to W1:
> We sincerely appreciate your question. Actually, the approach of applying a length penalty during RL training has already been included in our comparative experiments (Table 1 in the main text), denoted as "RL+length penalty" and "ShorterBetter".  Our results demonstrate that while these methods can reduce reasoning length to some extent, they often degrade accuracy, especially on challenging tasks, consistent with findings in papers [4] and [5].
>
> For instance (Table 1 in the main text), on the DeepSeek-R1-Distill-Qwen-7B model, the accuracy (Acc) of "RL + Length Penalty" and "ShorterBetter" drops below that of the Vanilla (51.9 vs. 53.3 vs. 55.4) on AIME 2024. In contrast, our proposed S-GRPO not only preserves accuracy but even improves it over the Vanilla (56.0 vs. 55.4), while still achieving a significant reduction in reasoning length.
> These results suggest that directly applying a length penalty is a relatively coarse control mechanism and may lead to underthinking on complex queries. In contrast, S-GRPO enables more nuanced control, achieving a better balance between efficiency and accuracy.
>
> ## W2:
> More discussion on early exiting in Related Works.
> ## Reply to W2:
> We sincerely appreciate the reviewer's suggestion to discuss the recommended works [1,2,3]. We will incorporate a detailed comparison in the Related Work section. These works indeed share the high-level concept of early-exit, but with fundamental differences in implementation: Compared with [2], while they focus on inter-layer early-exit (e.g., skipping transformer layers), our method operates at the reasoning-path level. Compared with [3], our approach eliminates the need for small model assistance by internalizing early-exit capabilities directly into the reasoning model through reinforcement learning.
> ## W3:
> How many GPUs were used for training?
> ## Reply to W3:
> Each training run was conducted on 64 GPUs with 80GB of memory. We have already provided details about the computational resources in Appendix B. We will further clarify this information in the main text.
>
> &nbsp;
> &nbsp;
>
> [1] Dynamic neural networks: A survey
>
> [2] LayerSkip: Enabling Early Exit Inference and Self-Speculative Decoding
>
> [3] A Stitch in Time Saves Nine: Small VLM is a Precise Guidance for Accelerating Large VLMs
>
> [4] Efficient RL Training for Reasoning Models via Length-Aware Optimization
>
> [5] Stable Reinforcement Learning for Efficient Reasoning

---

### Comment · Area_Chair_NhC6 · 2025-08-06

Dear reviewers,

This is a reminder that the end of author-reviewer discussion period is **Aug. 8**. Please do carefully read all other reviews and the author responses; and discuss openly with the authors, especially on your own questions that the authors addressed.

Best,
AC

---

### Note · Authors · 2025-08-12

We sincerely thank all reviewers for their constructive feedback.

Reviewers praised S-GRPO as a novel RL method that tackles overthinking in reasoning models, cutting reasoning length by 35.4%∼61.1% while boosting accuracy by 0.72%∼6.08%. They highlighted its elegant serial-group rollout with decaying rewards (65Nr, Rkde, N5Z5), strong experimental validation across benchmarks (6wUr, Rkde, N5Z5), and clear, high-quality presentation (N5Z5).

We also addressed several specific concerns raised in the reviews:

1. Both reviewers 65Nr and N5Z5 raised concerns about potential doubling of inference time in S-GRPO. We measured the actual cost (Qwen3-8B, rollout n=8) and found only around 20% additional overhead (457s → 550s) compared to GRPO, far below the “doubling” assumption. This is because the second rollout only generates the short conclusion, and the prefill of the second rollout is efficient. We also compared with simply doubling rollouts for "RL + Length Penalty" and confirmed that our two-phase design is more efficient and maintains accuracy.

2. For the prompt and truncation strategy question from reviewers 65Nr and N5Z5, we tested multiple templates and found that as long as `\n</think>\n\n` is present, the model terminates reasoning as intended, with similar accuracy. We also compared random truncation with semantic-based truncation (“Wait”, “Alternatively”), finding that the latter becomes unstable because the frequency of these semantic markers decreases. In contrast, random truncation remains robust, justifying our choice.

3. In response to N5Z5 and Rkde reviewers' request for broader evaluation, we added experiments on MMLU-Pro (general knowledge) and LiveCodeBench v5 (code). In both domains, S-GRPO maintains accuracy while substantially reducing output length, demonstrating its benefits across diverse domains.

4. Addressing reviewer 6wUr’s point on length penalties, we highlighted that while "RL + Length Penalty" or "ShorterBetter" can reduce reasoning length, they often harm accuracy on challenging tasks. In contrast, S-GRPO achieves length compression with preserved or even improved accuracy.

Through this rebuttal, our revisions have strengthened the paper's empirical validation of design details, efficiency, and generalization. We believe the enhanced paper, informed by the reviewers' expertise, will provide substantial value to the community in optimizing GRPO-like RL paradigms and addressing the pressing challenge of LLM efficiency.

---

### Decision · Program_Chairs · 2025-09-17

**Decision:**

Accept (poster)

**Comment:**

The paper introduces S-GRPO, an RL method that encourages early exit in chain-of-thought generation via serial truncation with decaying rewards. Compared to GRPO, it reduces reasoning length while maintaining or improving accuracy on multiple math/science benchmarks, and, per rebuttal, also on MMLU-Pro (general knowledge) and LiveCodeBench (code).

**Strengths:**

- Clear, simple extension to GRPO with practical impact.

- Consistent reductions in output length with preserved or improved accuracy.

- Solid experiments and ablations; results hold across models and tasks.

- Rebuttal provided broader evaluation and efficiency measurements (~20% overhead, not double).

**Weaknesses**

- Some reviewers felt simpler baselines (length penalties, prompts) could suffice; **novelty** seen as incremental.

- Main text focused on math/science; broader tasks only briefly covered.

- Added complexity in rollout design vs. simpler approaches.

--

The method is practically useful, well-validated, and integrates easily with existing GRPO pipelines. While not a fundamental paradigm shift or a clearly novel methodology, it delivers consistent, meaningful efficiency gains for reasoning LLMs. I recommend poster acceptance (not spotlight) given the incremental nature of the contribution.

**My quick note on discussions and rebuttal:**

- Training cost concern: Authors showed overhead is ~20%, not 2x.

- Prompt/truncation sensitivity: Tested multiple templates.

- Baseline adequacy: Authors compared against length penalties and showed accuracy drops; some reviewers remained unconvinced.

- Domain breadth: Added MMLU-Pro and LiveCodeBench, showing similar benefits.

- Two reviewers moved to clear acceptance, two stayed borderline; overall evidence supports acceptance.